# When Large Vision Language Models Meet Multimodal Sequential Recommendation: An Empirical Study

## Abstract

As multimedia content continues to grow on the Web, the integration of visual and textual data has become a crucial challenge for Web applications, particularly in recommendation systems. Large Vision Language Models (LVLMs) have demonstrated considerable potential in addressing this challenge across various tasks that require such multimodal integration. However, their application in multimodal sequential mmendation (MSR) has not been extensively studied, despite their potential to significantly enhance the performance of web-based multimodal recommendations. To bridge this gap, we introduce **MSRBench**, the first comprehensive benchmark designed to systematically evaluate different LVLM integration strategies in web-based recommendation scenarios. We benchmark three state-of-the-art LVLMs, i.e., GPT-4 Vision, GPT-4o, and Claude-3-Opus, on the next item prediction task using the constructed Amazon Review Plus dataset, which includes additional item descriptions generated by LVLMs. Our evaluation examines five integration strategies: using LVLMs as *recommender*, *item enhancer*, *reranker*, and various combinations of these roles. The benchmark results reveal that 1) using LVLMs as rerankers is the most effective strategy, significantly outperforming others that rely on LVLMs to directly generate recommendations or only enhance items; 2) GPT-4o consistently achieves the best performance across most scenarios, particularly when employed as a reranker; 3) the computational inefficiency of LVLMs presents a major barrier to their widespread adoption in real-time multimodal recommendation systems. Our codes and datasets will be made publicly available upon acceptance.

## CCS Concepts

• **Information systems** → **Recommender systems**.

## Keywords

Benchmark, Large Vision Language Model, Multimodal Recommendation

**ACM Reference Format:**

Anonymous Author(s). 2024. When Large Vision Language Models Meet Multimodal Sequential Recommendation: An Empirical Study. In *Proceedings of Make sure to enter the correct conference title from your rights confirmation emai (Conference acronym 'XX).* ACM, New York, NY, USA, 19 pages. https://doi.org/10.1145/nnnnnnn.nnnnnnn

## 1 Introduction

The explosive growth of multimedia content on the Web has fueled the need for more sophisticated recommendation systems that can handle diverse data modalities, such as images and text, to enhance user experiences. Multimodal Sequential Recommender Systems (MSRs), which integrate these multiple modalities, have witnessed a surge in popularity in recent years due to their superior capability in delivering more accurate and personalized web-based recommendations [28]. These systems typically encode each modality with a unique encoder and then employ complex fusion mechanisms to align the disparate data into a unified representation for the downstream recommendation task [28]. However, this shallow alignment approach may overlook the intricate correlations between different modalities, particularly when substantial differences exist across modalities [44]. As a result, the performance of these systems in integrating visual and textual features on the Web can be suboptimal.

Recently, the rapid development of Large Vision-Language Models (LVLMs) has profoundly influenced various Web applications, particularly those requiring the integration of visual and textual data, such as visual question answering [2, 3, 27], image captioning [38, 42], and cross-modal retrieval [21, 24]. The performance improvements in these fields can be largely attributed to LVLMs' strong ability to capture the complex relationships between images and text. However, how to effectively apply such strong ability of LVLMs to Web-based recommendation systems remains an open question. So far, only a few preliminary exploratory studies have been conducted: [47] and [29] are pioneer works that leverage prompt engineering to harness the recommendation capabilities of GPT-4 Vision [34]; [15] and [39] have also investigated the feasibility of directly employing LVLMs as multimodal recommenders. These studies do not use pre-trained LVLMs. Instead, they integrate vision encoders into language models, thereby endowing the language models with the capability to process visual signals. Different from the above works, [44] adopts LVLMs as a feature extractor and builds recommenders within traditional multimodal recommendation framework. Specifically, this work replaces separate encoders with pretrained LVLMs, demonstrating their potential to deeply align multimodal data and overcome the limitations of traditional shallow alignment methods.

Despite these advancements, the aforementioned studies have primarily focused on exploring the effectiveness of applying LVLMs in recommendation scenarios using a single integration approach. However, there is a lack of a comprehensive performance evaluation of different integration approaches (e.g., LVLMs as a recommender, LVLMs as a reranker) for the same task within the context of Web applications. In other words, we still do not fully understand the performance disparities among these various integration strategies when applied to Web-based recommendation systems. Thus, there is an urgent need for a systematic benchmark to thoroughly assess the different integration strategies of LVLMs in multimodal sequential recommendation scenarios, facilitating more informed decisions in model selection and deployment.

To fill this gap, we construct MSRBench, the first benchmark that comprehensively evaluates different strategies for integrating

Conference acronym 'XX, June 03–05, 2018, Woodstock, NY
2024. ACM ISBN 978-1-4503-XXXX-X/18/06
https://doi.org/10.1145/nnnnnnn.nnnnnnn

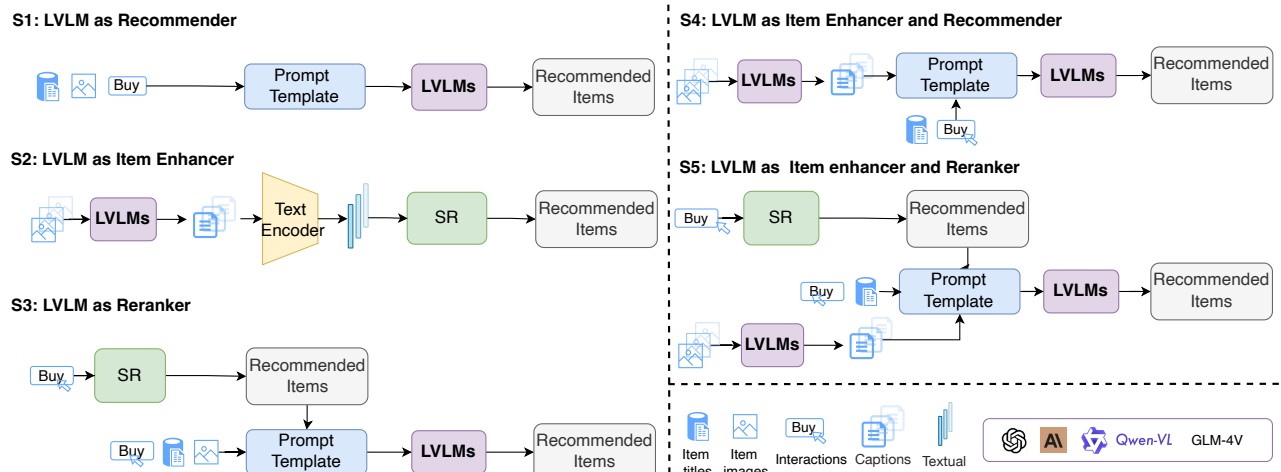

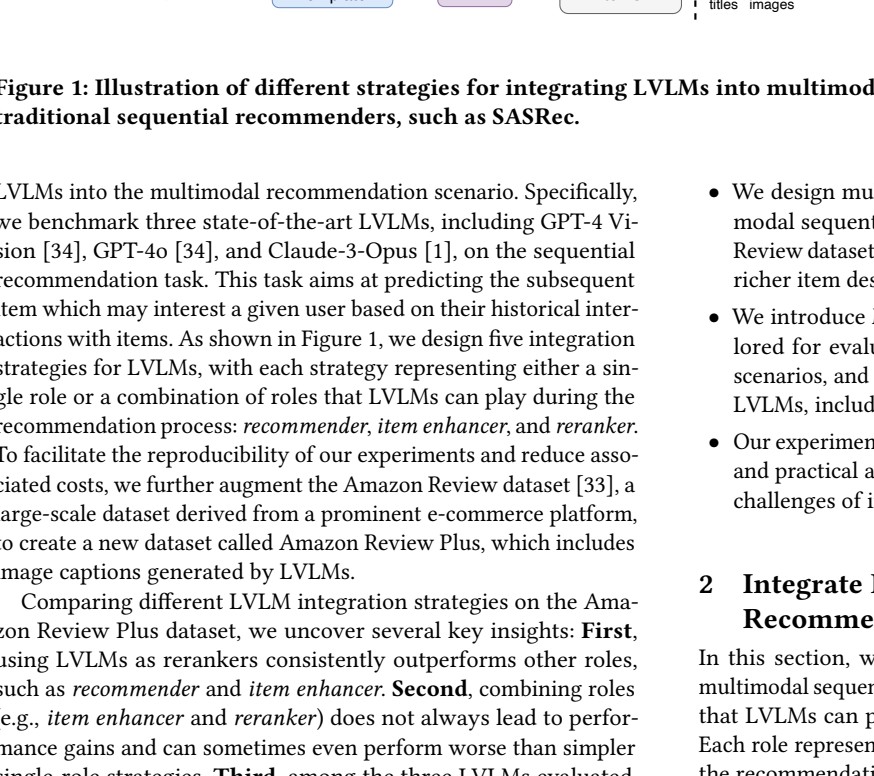

**Figure 1: Illustration of different strategies for integrating LVLMs into multimodal sequential recommendation. SR denotes traditional sequential recommenders, such as SASRec.**

LVLMs into the multimodal recommendation scenario. Specifically, we benchmark three state-of-the-art LVLMs, including GPT-4 Vision [34], GPT-4o [34], and Claude-3-Opus [1], on the sequential recommendation task. This task aims at predicting the subsequent item which may interest a given user based on their historical interactions with items. As shown in Figure 1, we design five integration strategies for LVLMs, with each strategy representing either a single role or a combination of roles that LVLMs can play during the recommendation process: *recommender*, *item enhancer*, and *reranker*. To facilitate the reproducibility of our experiments and reduce associated costs, we further augment the Amazon Review dataset [33], a large-scale dataset derived from a prominent e-commerce platform, to create a new dataset called Amazon Review Plus, which includes image captions generated by LVLMs.

Comparing different LVLM integration strategies on the Amazon Review Plus dataset, we uncover several key insights: **First**, using LVLMs as rerankers consistently outperforms other roles, such as *recommender* and *item enhancer*. **Second**, combining roles (e.g., *item enhancer* and *reranker*) does not always lead to performance gains and can sometimes even perform worse than simpler single-role strategies. **Third**, among the three LVLMs evaluated, GPT-4o demonstrates superior performance across most strategies. **Finally**, and most importantly, despite these performance gains, the computational inefficiency of LVLMs, especially in more complex strategies, remains a significant challenge for real-time deployment in industrial systems. We hope these insights can deepen the understanding of how different LVLM integration strategies impact recommendation performance, highlight the most promising approaches for leveraging LVLMs, and validate both their utility and the challenges they currently present. Furthermore, we hope MSRBench can guide future improvements in model design and integration techniques, and encourage further exploration of LVLMs in diverse web-based recommendation scenarios.

To summarize, our contributions are threefold:

- We design multiple strategies for leveraging LVLMs in multimodal sequential recommendation and augment the Amazon Review dataset by creating Amazon Review Plus, which includes richer item descriptions to enable more flexible item modeling.

- We introduce MSRBench, the first benchmark specifically tailored for evaluating LVLMs in multimodal recommendation scenarios, and conduct extensive evaluations of state-of-the-art LVLMs, including GPT-4V, GPT-4o, and Claude-3-Opus.

- Our experimental results offer clear guidance for future research and practical applications, outlining both the potential and the challenges of integrating LVLMs into recommender systems.

## 2 Integrate LVLM into Mutimodal Sequential Recommendation

In this section, we explore how LVLMs can be integrated into multimodal sequential recommendation, focusing on three key roles that LVLMs can play: *recommender*, *item enhancer*, and *reranker*. Each role represents a different way in which LVLMs contribute to the recommendation process:

- **Recommender**: The LVLM directly generates recommendations by processing multimodal data (e.g., item images and titles) to predict the items the user is most likely to be interested in.

- **Item Enhancer**: The LVLM enriches item descriptions by converting visual information into textual form, such as generating image captions to improve item metadata.

- **Reranker**: The LVLM refines the output of a traditional sequential recommender by reevaluating the recommended items and adjusting their order based on multimodal input.

We propose five distinct integration strategies based on these roles, as illustrated in Figure 1. The first three strategies (S1, S2, S3) focus on LVLMs taking on a single role, while the latter two strategies (S4, S5) explore whether combining different roles can

lead to performance improvements. We describe the details of each strategy as follows:

**Strategy 1 (S1): LVLM as a direct recommender.** This strategy inputs both the images and titles of items previously interacted with by the user into the LVLM, enabling it to identify the user's preferences and generate personalized recommendations. Specifically, the images of all previously interacted items are concatenated into a single image, arranged in chronological order. As shown in Figure 6 in the appendix, the prompt template guides the LVLM on how to interpret the concatenated image and link it to the corresponding item titles. In this setup, the LVLM simultaneously leverages both visual and textual information to generate its recommendations. Additionally, the prompt imposes format constraints, requiring the model to output the recommendation list in a structured format (e.g., JSON) and provide explanations for its recommendations.

**Strategy 2 (S2): LVLM as an item enhancer.** In this strategy, we leverage LVLMs to transform visual information into textual form[1]. Specifically, the LVLM generates image captions that describe the content of item images, thereby enriching the item's textual metadata. To assess the effectiveness of this strategy, we use BERT [9] to encode the enhanced textual data (i.e., the combination of titles and image captions) and obtain semantic representations of the items. These representations are then used to initialize the embedding table of traditional sequential recommendation models, such as SASRec [19], thereby incorporating prior knowledge from the LVLM-enhanced data into the recommendation models.

**Strategy 3 (S3): LVLM as a reranker.** In this strategy, we design a prompt template that can be used for reranking recommendation lists from other recommendation models, such as SASRec. As shown in Figure 6, the template prompts the LVLM to reassess the relevance of the recommended items based on both the titles and images of previously interacted items. The LVLM then outputs a reranked recommendation list, prioritizing items that better align with the user's preferences.

**Strategy 4 (S4): LVLM as both item enhancer and recommender.** This strategy first utilizes an LVLM to generate captions for each item's image. These captions, combined with the item titles, are then used to represent the items. As shown in Figure 6, this textual representation is injected into the prompt template from S1, replacing the original image input. The updated prompt is processed by the LVLM, and the final recommendations are derived from its output. Notably, in S4, item images are not directly fed into the LVLM for recommendations. Instead, the images are transformed into captions, and the recommendations are generated based on these captions and the associated titles.

**Strategy 5 (S5): LVLM as both item enhancer and reranker.** This strategy first utilizes LVLM to obtain image captions. These captions, combined with item titles, are then used to rerank the recommendation list produced by other models. Similar to S4, this approach does not use the raw image data for reranking; instead, it leverages the transformed textual information. For the prompt template used in S5, please refer to Figure 6 in the appendix.

---

[1]we discuss why the reverse approach (i.e., converting textual modality into visual modality) is not considered in Appendix A.5

## 3 MSRBench

In this section, we conduct comprehensive experiments to answer the following key research questions:

- **RQ1:** How do LVLMs perform when integrated into multimodal sequential recommendation systems in various roles?
- **RQ2:** How do different item modalities and different image input modes (e.g., multiple images vs. concatenated images) affect recommendation performance?
- **RQ3:** Can LVLMs, when used as rerankers, consistently enhance the performance of different traditional sequential recommenders?
- **RQ4:** Which LVLM integration strategy offers the best trade-off between computational efficiency and recommendation accuracy?

### 3.1 Experimental Settings

*3.1.1 Model Selection.* The primary goal of MSRBench is to investigate the impact of LVLMs in different roles within a sequential recommendation process. To this end, we select three state-of-the-art commercial LVLMs: GPT-4V, GPT-4o, and Claude-3-Opus.These models are used to evaluate the five integration strategies introduced in Section 2. We also explored several open-source LVLMs, including Qwen-VL [2] and GLM-4V [16]. Unfortunately, these open-source models exhibited poor instruction-following capabilities, with outputs that were either difficult to parse or plagued by severe hallucination issues. Therefore, we exclude them in our following experiments.

For our recommendation baselines, we categorize the models into three groups: classical sequential recommendation methods, collaborative multimodal recommendation methods, and state-of-the-art multimodal sequential recommendation methods. The classical sequential recommendation methods include: 1) **Pop**, which ranks items based solely on their popularity, recommending the most popular items to users first, and 2) **SASRec**[19], an ID-based sequential recommendation model that uses self-attention mechanisms to capture user-item interaction patterns, enabling long-term preference modeling. The collaborative multimodal recommendation methods consist of: 1) **MMGCN**[40], a graph-based model that leverages multimodal features and graph convolutional networks to enhance recommendation performance, 2) **FREEDOM**[49], which freezes the item-item graph and denoises the user-item graph for efficient and accurate multimodal recommendations, and 3) **BM3**[50], a self-supervised model that uses dropout to generate contrastive views without the need for negative samples, improving the robustness of the recommendations. Finally, the state-of-the-art multimodal sequential recommendation methods include: 1) **MoRec**[45], which integrates multimodal features to enhance recommendation accuracy and is evaluated using three input configurations: text-only (*MoRec (T)*), image-only (*MoRec (I)*), and a combination of text and image features (*MoRec (T+I)*), and 2) **IISAN**[11], a cutting-edge method that employs a decoupled parameter-efficient fine-tuning (PEFT) framework, leveraging both intra- and inter-modal adaptation to improve training efficiency and memory usage while maintaining strong recommendation performance.

**Table 1: Performance comparison between different strategies for integrating LVLM into multimodal recommendation. T and I denote _title_ and _image_ of items, respectively. Bold text indicates the highest score, and underlined text indicates the second highest score. "*" denotes statistical significance for $p < 0.01$ between the best method and all other methods, based on a paired $t$-test. All results are presented as percentages to ensure clarity and ease of reading.**

| Method | Strategy | Beauty | | | Sports | | | Toys | | | Clothing | | |
|---|---|---|---|---|---|---|---|---|---|---|---|---|---|
| | | H@1 | H@5 | N@5 | H@1 | H@5 | N@5 | H@1 | H@5 | N@5 | H@1 | H@5 | N@5 |
| Random | - | 4.75 | 16.50 | 10.57 | 3.75 | 14.75 | 9.01 | 5.25 | 18.25 | 11.54 | 4.50 | 15.00 | 9.68 |
| Pop | - | 14.00 | 41.00 | 28.00 | 14.50 | 38.00 | 26.81 | 10.25 | 33.50 | 21.86 | 15.25 | 42.00 | 28.41 |
| SASRec | - | 26.25 | 50.50 | 38.09 | 18.25 | 50.00 | 34.01 | 22.25 | 43.00 | 32.54 | 13.75 | 36.50 | 25.12 |
| MMGCN | - | 22.50 | 49.75 | 36.70 | 20.00 | 54.25 | 37.40 | 18.75 | 45.25 | 32.55 | 16.25 | 42.00 | 29.29 |
| FREEDOM | - | 33.00 | 59.25 | 46.89 | 33.75* | 64.25 | 49.53 | 32.50 | 61.00 | 47.64 | 26.75 | 49.75 | 38.80 |
| BM3 | - | 29.00 | 54.75 | 42.91 | 29.75 | 62.50 | 47.11 | 25.00 | 53.50 | 40.09 | 22.25 | 50.00 | 36.21 |
| IISAN | - | 9.92 | 29.48 | 19.60 | 12.70 | 37.64 | 24.97 | 11.57 | 31.57 | 21.68 | 9.48 | 30.02 | 19.52 |
| MoRec (T) | - | 31.00 | 58.00 | 45.51 | 27.25 | 62.50 | 45.62 | 28.00 | 60.00 | 44.70 | 24.50 | 55.50 | 40.12 |
| MoRec (I) | - | 34.00 | 58.25 | 47.08 | 23.50 | 63.00 | 43.94 | 30.75 | 59.50 | 45.74 | 24.50 | 56.75 | 40.81 |
| MoRec (T+I) | - | 33.00 | 61.50 | 47.92 | 28.75 | 67.50* | 49.23 | 33.25 | 64.75* | 50.03 | 27.00 | 59.75 | 43.92 |
| GPT-4V | S1 | 23.74 | 46.46 | 34.98 | 23.00 | 54.25 | 38.77 | 28.50 | 49.00 | 39.41 | 20.41 | 47.70 | 34.49 |
| | S2 | 30.75 | 60.75 | 46.46 | 28.50 | 62.25 | 45.77 | 27.75 | 60.00 | 44.91 | 22.50 | 59.50 | 41.45 |
| | S3 | 31.71 | 57.54 | 45.30 | 32.06 | 63.61 | 48.37 | 32.91 | 59.80 | 46.84 | 28.17 | 57.36 | 43.04 |
| | S4 | 22.86 | 48.74 | 35.84 | 21.91 | 51.64 | 37.25 | 29.32 | 53.38 | 41.86 | 20.25 | 48.86 | 35.29 |
| | S5 | 32.15 | 55.44 | 44.09 | 31.38 | 62.76 | 47.12 | 32.04 | 57.88 | 45.40 | 22.28 | 53.58 | 38.45 |
| GPT-4o | S1 | 23.37 | 49.00 | 36.84 | 26.50 | 56.00 | 41.36 | 30.00 | 55.25 | 43.11 | 22.31 | 51.88 | 37.42 |
| | S2 | 30.75 | 60.00 | 46.03 | 26.50 | 62.50 | 44.98 | 29.75 | 59.25 | 45.00 | 26.50 | 53.25 | 40.20 |
| | S3 | **38.85*** | **61.90*** | **50.66*** | 30.83 | 65.41 | 49.01 | 37.50 | **64.75*** | 52.14 | **32.83*** | **61.40*** | **47.63*** |
| | S4 | 25.25 | 48.25 | 37.30 | 23.25 | 56.75 | 40.79 | 32.75 | 55.50 | 44.71 | 23.31 | 48.12 | 35.90 |
| | S5 | 38.00 | 59.50 | 49.17 | 33.00 | 65.00 | **49.71*** | **40.50*** | 64.00 | **52.84*** | 29.57 | 58.90 | 45.28 |
| Claude 3-Opus | S1 | 22.42 | 38.79 | 30.95 | 21.25 | 52.00 | 37.13 | 29.00 | 51.25 | 40.76 | 19.35 | 48.74 | 34.28 |
| | S2 | 31.00 | 59.75 | 46.10 | 26.00 | 62.00 | 44.82 | 29.25 | 59.25 | 44.94 | 21.50 | 60.00 | 41.17 |
| | S3 | 30.40 | 53.77 | 42.06 | 26.75 | 60.00 | 43.96 | 32.75 | 55.75 | 44.98 | 26.82 | 55.64 | 41.64 |
| | S4 | 26.00 | 51.75 | 39.15 | 22.61 | 57.04 | 39.76 | 27.25 | 52.25 | 40.78 | 22.56 | 51.13 | 37.67 |
| | S5 | 30.75 | 53.50 | 42.13 | 24.56 | 62.66 | 44.16 | 29.00 | 53.50 | 41.63 | 26.07 | 55.89 | 41.51 |

Note that we deliberately exclude models [22, 39] with complex architectures or intricate fusion mechanisms, as they could obscure the isolated effects of LVLM integration. Therefore, we only choose these seven models to maintain a controlled environment that allows us to provide clear insights into the specific benefits LVLMs bring to sequential recommendation systems.

_3.1.2 Dataset._ In this study, we conduct experiments using the Amazon Review dataset[2][32], which is widely adopted in sequential recommendation research[4, 11, 15, 25]. This dataset is particularly well-suited for our benchmark for two key reasons. First, it offers a rich collection of user interactions along with textual and visual product information. Second, it reflects real-world e-commerce scenarios, where users frequently rely on both product images and descriptions during their decision-making processes. Following prior works [15, 19, 48], we focus on four categories: beauty, sports, toys, and clothing, as they represent diverse consumer goods with distinct characteristics and interaction patterns.

In addition, we extend the original Amazon Review dataset to create the **Amazon Review Plus** dataset[3]. The motivation behind this extension is that in Strategy 2 (S2) and Strategy 5 (S5), LVLMs are used to enhance items by converting visual information into textual descriptions, specifically by generating image captions to enrich item metadata. The generation of these captions required approximately 192,480 API calls, covering 64,160 images across the four categories, each processed by three state-of-the-art LVLMs. This process took approximately 7 days, with API costs exceeding $2,000. Beyond financial costs, we faced challenges such as API rate limits, which required careful scheduling and batching to efficiently complete the task. To support the research community and mitigate these costs and challenges, we plan to make all generated image captions publicly available by integrating them into the original Amazon Review dataset, forming the Amazon Review Plus dataset. This resource offers a valuable foundation for future multimodal recommendation research, enabling others to leverage LVLM-enhanced item descriptions to improve recommendation performance. Following common practice, we apply the 5-core filtering, which retains only users and items with at least five interactions. Additionally, we employ the leave-one-out strategy to split the dataset: for each user, the last interacted item is used for testing, the second-to-last for validation, and the rest for training. Detailed statistics of the Amazon Review Plus dataset are presented in Appendix Table 4. Further analysis of the image captions, including quality evaluations, can be found in the Appendix A.7.

---

[2]http://jmcauley.ucsd.edu/data/amazon/
[3]The dataset consists of the four aforementioned categories

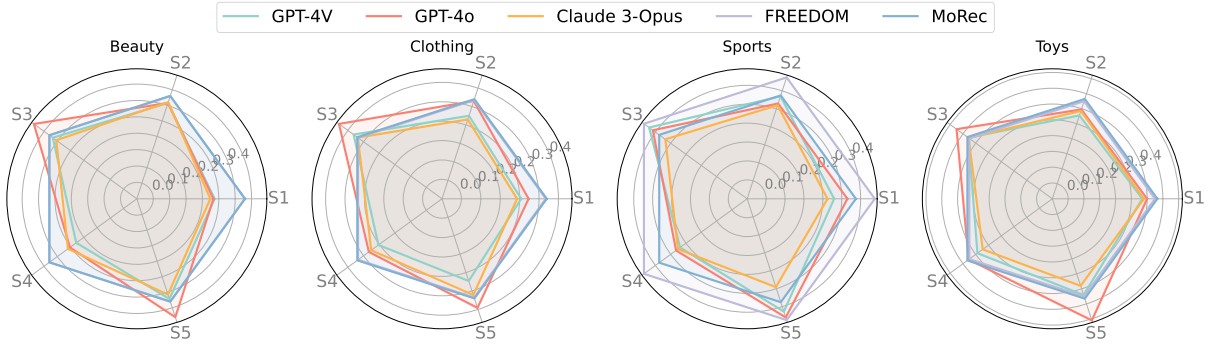

**Figure 2: Performance (H@1) comparison of GPT-4V, GPT-4o and Claude-3-Opus under different integration strategies.**

*3.1.3 Evaluation Metrics.* We adopt two widely-used metrics, top-k Hit Ratio (H@k) and top-k Normalized Discounted Cumulative Gain (N@k), to evaluate the recommendation performance of LVLMs under the five strategies.

When using LVLMs as recommenders or rerankers, the candidate items must be included in the input prompt for the model to rank them[4]. Given the prompt length limitations and high inference costs, evaluating with all unrated items (i.e., full-item ranking) is impractical. Therefore, following previous studies [8, 25, 30], we evaluate the models by ranking a subset of candidate items. This subset consists of 1 target item and 29 randomly sampled negative items. We do not use the commonly applied 99 negative samples because our experiments on the *Beauty* category show that, while absolute metrics differ, the relative model rankings remain consistent, and the overall performance trends and conclusions stay unchanged. Using 29 samples also significantly reduces prompt length, which lowers inference time and costs, making it a more efficient and practical choice for large-scale applications without compromising result reliability.

*3.1.4 Implementation Details.* For the five integration strategies, we design task-specific prompts for LVLMs to handle the multimodal sequential recommendation task (details shown in Fig 6). In S2, we use the bert-base-uncased[5] as the text encoder, with a text embedding dimension of 768. For all LVLM-related experiments, we set the temperature of LVLMs to 0 to ensure reproducibility. Following [12, 41], we set the maximum user interaction sequence length to 10 for inputs to the model. For the recommendation baselines, we implement SASRec and MoRec using the code[6] provided by [45] with default hyperparameters: the number of transformer blocks and attention heads are both set to 2, embedding dimensions to 512, and the dropout ratio to 0.1. To ensure fair performance reporting, we search for the optimal batch size from 32, 64 and the best learning rate from $1e-5, 5e-5, 1e-4$, ultimately setting the batch size to 64 and the learning rate to 1e-4 for all experiments. For MMGCN, FREEDOM, and BM3, we use the implementations provided in MMRec[7]. For IISAN, we utilize the code[8] released by

the authors and perform a grid search over the hyperparameters as recommended in the original paper, reporting the best results. Due to space limitations, more detailed implementation settings of these baselines can be found in the Appendix A.6.

## 3.2 Overall Performance (RQ1)

Table 1 demonstrates the performance of various LVLMs when evaluated using the five proposed strategies on the four categories of Amazon Review Plus dataset. We make the following observations:
**LVLM as a reranker (S3) is the most effective single-role strategy, outperforming both the recommender (S1) and item enhancer (S2) strategies.** The reranking mechanism allows LVLMs to refine existing recommendation lists, effectively leveraging multimodal data for better item ordering. For example, in the beauty category, GPT-4o as a reranker (S3) achieves an H@1 of **38.85%** and N@5 of **50.66%**, significantly outperforming its performance as a recommender (S1), where H@1 was **23.37%**. This suggests that LVLMs perform best when refining pre-generated recommendations, rather than directly generating recommendations (S1) or solely enriching item descriptions (S2), particularly when handling complex multimodal data.
**Combination strategies (S4 and S5) show mixed results when compared to their corresponding single-role strategies (S1, S2, and S3), and neither consistently outperforms them.** S5, which combines item enhancement and reranking, often improves upon the item enhancer (S2) and reranker (S3) strategies, but the gains are not always substantial. For instance, in the toys category, S5 achieves an H@1 of 40.50%, which is an improvement over S2's 30.75%, but only marginally better than S3's 37.50%. In contrast, in the beauty category, S3 outperforms S5, achieving an H@1 of 38.85% compared to S5's 38.00%. Similarly, S4, which combines item enhancement and recommender roles, fails to consistently outperform its single-role counterparts. In the sports category, S4 achieves an H@1 of 23.25%, which is lower than both S1 (26.50%) and S2 (30.75%). These results suggest that while combining roles (as in S4 and S5) can occasionally offer improvements, particularly for S5, the more straightforward single-role strategies often deliver comparable or even better results.
**GPT-4o consistently outperforms GPT-4V and Claude 3-Opus across all strategies, especially as a reranker (S3).** GPT-4o demonstrates superior handling of multimodal data, achieving the highest scores with S3 across categories such as beauty, where it

---

[4]As shown in [25], directly performing sequential recommendation without including candidate items leads to severe hallucination
[5]https://huggingface.co/google-bert/bert-base-uncased
[6]https://github.com/westlake-repl/IDvs.MoRec
[7]https://github.com/enoche/MMRec
[8]https://github.com/GAIR-Lab/IISAN

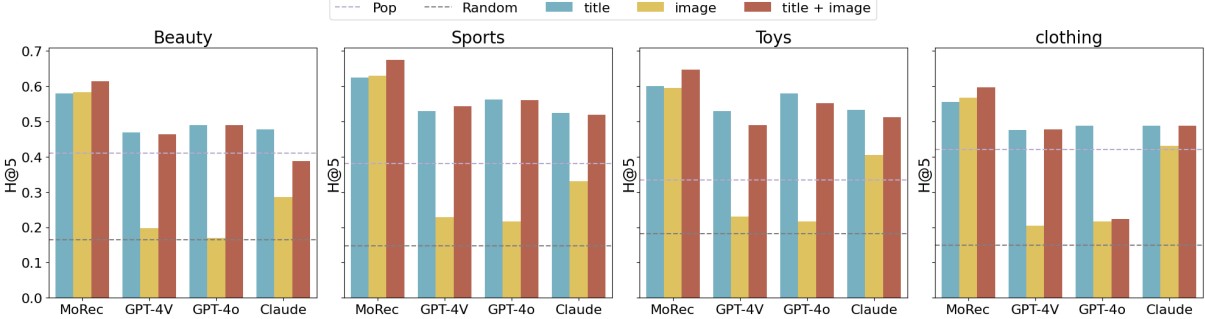

**Figure 3: Impact of different item modalities (*title only, image only, title + image*) on recommendation performance.**

reaches an H@1 of **38.85%**, and clothing, where it reaches **32.83%**. GPT-4V, though stable, underperforms GPT-4o in more complex strategies, such as S4 and S5, while Claude 3-Opus struggles the most, particularly in the direct recommender strategy (S1), where its ability to generate accurate recommendations from raw multimodal data is limited.

**Comparison among recommendation baselines.** Classical SR methods like Pop and SASRec perform weaker, with Pop achieving only 14.00% H@1 in beauty and 14.50% in sports. These models, which rely on popularity-based ranking or simple sequential modeling, struggle to compete with more advanced multimodal approaches. Furthermore, we select the best-performing models from the baseline groups, namely FREEDOM and MoRec (T+I), and compare them with our proposed LVLM-based methods in Figure 2. We observe that both S3 and S5 applied to GPT-4o outperform these baselines in most cases, particularly in the beauty, clothing, and toys categories. This highlights the ability of these two strategies to more effectively leverage multimodal data, leading to improved recommendation accuracy.

### 3.3 Impact of Different Item Modalities (RQ2)

This section explores the influence of different item modalities when using LVLMs as a recommender (S1). As shown in Figure 3, item titles consistently emerge as the most critical information for generating accurate recommendations, while images alone lead to significantly poorer performance. For instance, when relying solely on images, GPT-4V and GPT-4o produce results close to random selection, and although Claude 3 Opus performs better, it still falls short of the Pop baseline. This suggests that LVLMs cannot yet depend on image data alone for reliable recommendations, possibly due to noise or ambiguity in the images. Additionally, combining titles and images does not improve performance in S1, indicating that current LVLMs struggle to process raw image data effectively in this context. As highlighted in the Sec. 3.2 , transforming images into textual descriptions or using images as auxiliary signals for reranking items remains the most effective way for LVLMs to enhance recommendation accuracy.

When leveraging LVLMs as recommenders or re-rankers, one essential step is to input images of items from users' historical interactions into the LVLM. We explore two different input modes: (Mode 1) concatenating images into a single composite image, and

(Mode 2) inputting images individually. In both modes, the images are arranged in chronological order based on the interaction timestamp. For instance, in Mode 1, the leftmost item in the concatenated image represents the first interaction. Our experiments using GPT-4V on the beauty dataset reveal that Mode 1 consistently outperforms Mode 2 across key metrics. Specifically, Mode 1 achieves an H@1 of 23.74%, whereas Mode 2 results in lower H@1 scores of 22.86%. This indicates that concatenating images into a single composite (Mode 1) helps the model capture the relationships between items more effectively, thereby reflecting the user's preferences more accurately.

### 3.4 In-depth Analysis of LVLM as Reranker (RQ3)

In this section, we investigate whether LVLMs, when used as rerankers, can enhance the performance of different sequential recommendation models (referred to as backbones for simplicity), such as SASRec and MoRec. Specifically, we first generate the initial recommendation lists from both backbone, and then apply two reranking strategies: S3 (LVLM as reranker) and S5 (LVLM as both item enhancer and reranker). Due to space constraints, as shown in Figure 4, we present results for GPT-4o only. We observe that for both SASRec and MoRec, S3 and S5 can consistently improve their recommendation performances. Notably, the S3 strategy, where the LVLM operates purely as a reranker by directly processing product images, performs best in most cases. In contrast, S5 first converts images into textual descriptions before reranking, which can lead to information loss or the introduction of irrelevant details, diminishing its effectiveness in certain scenarios. These results highlight the combination of textual and visual data, alongside effective reranking strategies, provides a promising avenue for enhancing the overall performance of recommendation systems.

In addition to improving accuracy, both S3 and S5 offer explainable recommendations by providing detailed explanations alongside the results. These explanations help users understand the reasoning behind the recommendations, which is key to enhancing user satisfaction. Figure 5 showcases GPT-4o functioning as an (item enhancer and) reranker, compared to the traditional SASRec system. In this instance, SASRec incorrectly suggests a finger puppet, unrelated to the user's history with dolls, and offers no explanation for this error. In contrast, S3 highlights the user's preference for

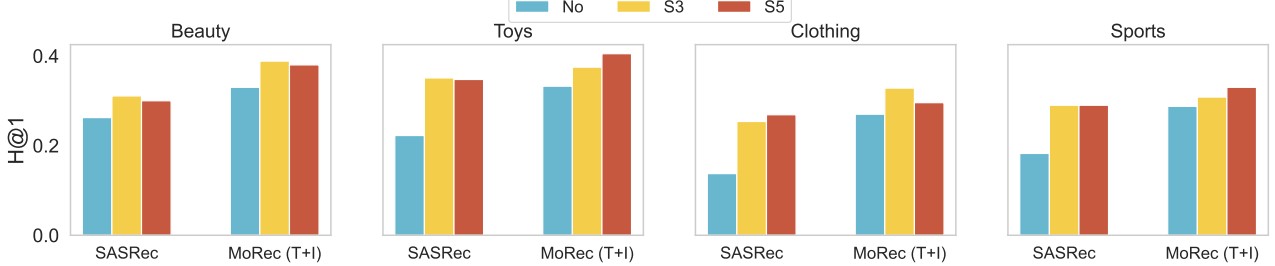

**Figure 4: Performance comparison between different LVLM-based reranking strategies (S3 and S5) using two different recommendation backbones. In this analysis, GPT-4o is used as the LVLM.**

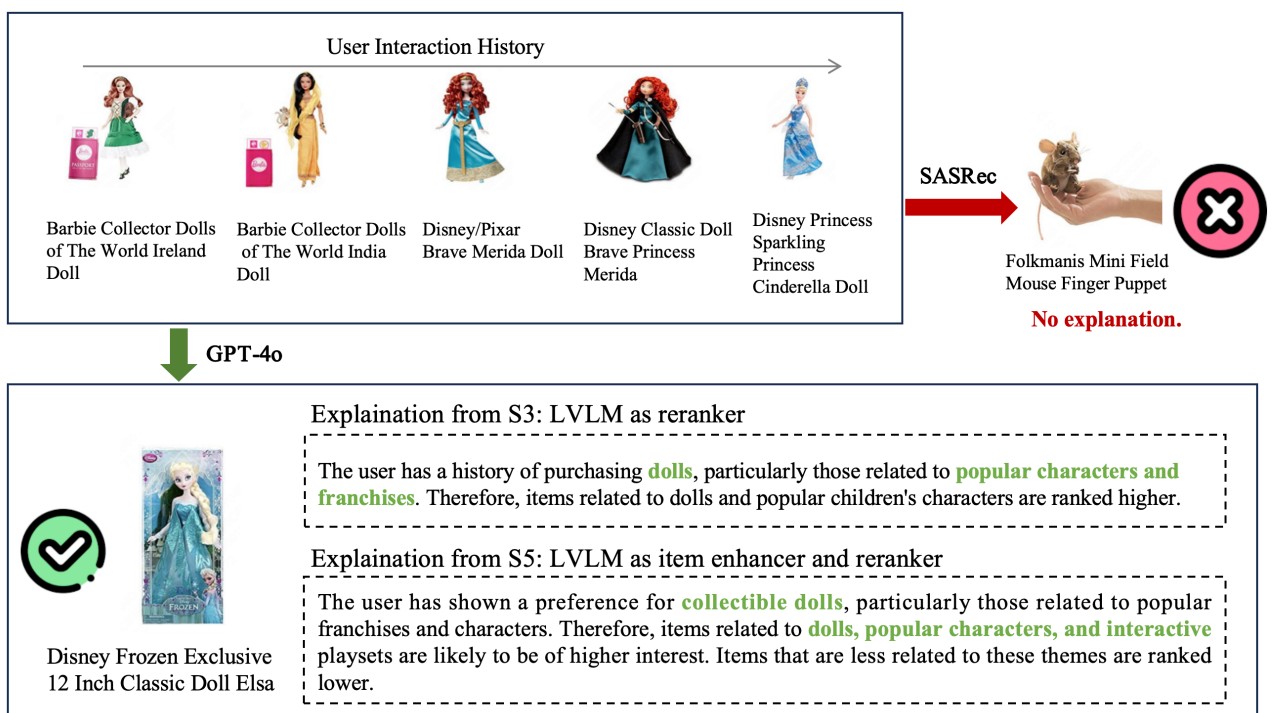

**Figure 5: Case study between GPT-4o and SASRec. GPT-4o can provide detailed explainations for its recommendation results.**

dolls related to popular characters, while S5 enriches the explanation by factoring in collectible dolls and interactive playsets. This added depth improves transparency and personalization, ultimately increasing user trust and satisfaction.

## 3.5 Efficiency Comparison (RQ4)

This section investigates the computational efficiency of different LVLM integration strategies and all baseline models in multimodal sequential recommendation. We conduct experiments on the Beauty category of the Amazon Review Plus dataset, with batch size unified to 64 for all baselines to ensure fair comparison. Table 2 summarizes the results, and our key observations are as follows:

**SASRec and MoRec demonstrate the fastest training speed, outperforming collaborative multimodal recommendation**

**methods.** SASRec completes an epoch in 5.4 seconds, while MoRec (T+I) takes 11 seconds per epoch. In contrast, collaborative models such as FREEDOM and BM3 exhibit similar training times, around 401.89 and 411.45 seconds per epoch, respectively. MMGCN is the slowest, requiring 674.99 seconds per epoch. The slower training time of MMGCN can likely be attributed to its graph-based architecture, which adds computational complexity. Despite these differences, all baseline models maintain high inference speeds, with most achieving 0.0025 seconds per user.

**The computational inefficiency of LVLMs poses a major barrier to their widespread adoption in multimodal recommendation systems.** Among the LVLM integration strategies, S2 (item enhancer) stands out as the most practical and feasible approach for industrial deployment. It achieves a training time of 5.9

**Table 2: Comparison of computational efficiency in different LVLM integration strategies. Note that training time for S1 to S5 (except S2) is not applicable (N/A) as these methods are prompting-based.**

| Model | Train Time (s/epoch) | Inference Time (s/user) |
|---|---|---|
| SASRec | 5.4 | 0.0025 |
| MMGCN | 674.99 | 0.0030 |
| FREEDOM | 401.89 | 0.0029 |
| BM3 | 411.45 | 0.0025 |
| IISAN | 363.00 | 0.0004 |
| MoRec (T) | 5.9 | 0.0025 |
| MoRec (I) | 9.75 | 0.0025 |
| MoRec (T+I) | 11 | 0.0025 |
| S1: Recommender | | |
| GPT-4V | N/A | 41.7488 |
| GPT-4o | N/A | 25.8993 |
| Claude-3-Opus | N/A | 24.9818 |
| S2: Item Enhancer | | |
| GPT-4V/o, Claude-3-Opus | 5.9 | 0.0025 |
| S3: Reranker | | |
| GPT-4V | N/A | 42.4901 |
| GPT-4o | N/A | 24.4900 |
| Claude-3-Opus | N/A | 24.7115 |
| S4: Enhancer + Recommender | | |
| GPT-4V | N/A | 29.8392 |
| GPT-4o | N/A | 18.3205 |
| Claude-3-Opus | N/A | 23.5321 |
| S5: Enhancer + Reranker | | |
| GPT-4V | N/A | 35.4451 |
| GPT-4o | N/A | 28.8022 |
| Claude-3-Opus | N/A | 21.8895 |

seconds per epoch and an inference time of 0.0025 seconds per user, making it the most computationally efficient LVLM integration strategy. Additionally, S2 performs comparably to current SOTA multimodal sequential recommendation methods across multiple metrics, positioning it as a promising candidate for further exploration and optimization. More complex strategies, such as S3 to S5, which combine multiple roles, incur significantly higher inference costs. For example, S3, which uses LVLMs as rerankers, achieves the best recommendation performance but requires 42.49 seconds per user for inference with GPT-4V, far exceeding the inference times of baseline models. This presents a significant challenge for the application of LVLMs in real-time industrial systems, where low-latency requirements are critical.

## 4 Related Work

**Multimodal Sequential Recommendation.** multimodal sequential recommendation (MSR) integrates various data modalities (e.g., text and image) to better capture user preferences and improve recommendation accuracy. Early work like MV-RNN [7] introduced a multi-view recurrent neural network that dynamically fused multimodal features, while MML [35] leveraged meta-learning to address the cold-start problem using multimodal side information. Building on these advances, MMSR [17] introduced a graph-based model with dual attention mechanisms to fuse multimodal features from both homogeneous and heterogeneous user-item interactions. MMMLP [22] demonstrated the effectiveness of simpler, purely MLP-based architectures by efficiently fusing multimodal sequences

for large-scale recommendation tasks. Despite the recent success of LVLMs in various NLP tasks, only a few works have explored their application in the MSR domain. For instance, MLLM-MSR [43] and Rec-GPT4V [29] leverage LVLMs to summarize item images and combine them with titles to model user preferences, while UniMP [39] proposes a general framework for personalized recommendations. However, these methods apply LVLMs in a single integration manner without comprehensively evaluating their performance across different strategies. To address this gap, our work introduces MSRBench, the first benchmark designed to systematically explore the impact of different LVLMs, integration methods, and input modalities on recommendation performance. By investigating LVLMs in various roles, our analysis can provide new insights into performance differences across strategies, offering a more thorough evaluation than previous approaches.

**Large Language Models in Recommendation.** Large Language Models (LLMs) have revolutionized AI research by pushing the boundaries of natural language understanding and generation. Commercial close-sourced models like GPT [34] and Claude [1] demonstrate the impressive ability of coherent text generation and instruction following. The open-source counterparts, *e.g.,* LLaMA [37] and Vicuna [5], provide transparency into the model architecture and training details, allowing more flexible and customized development. For the recommendation side, several works [23, 25, 26, 31, 46] prompt or finetune LLMs to adapt them for recommendation. As the pioneer attempt, Liu *et.al* [25] prompt and evaluate ChatGPT's performance on the recommendation scenarios in a training-free way. LlamaRec [46] resorts to the open-source LLMs and renders a real-time recommendation to streamline the autoregressive generation during inference time. Since the recommendation system is essentially a multimodal system requiring the understanding of both textual and visual data, our paper focuses on investing multimodal LLMs for recommendations to uncover and unleash their ability for the effective and personalized recommendation.

## 5 Conclusion and Future Work

In this work, we introduced **MSRBench**, a comprehensive benchmark designed to evaluate different integration strategies of LVLMs in multimodal recommendation systems. By systematically benchmarking state-of-the-art LVLMs, including GPT-4 Vision, GPT-4o, and Claude-3-Opus, on the next item prediction task using the enhanced Amazon Review Plus dataset, we uncovered significant performance disparities among various integration approaches. Our findings highlight the most effective methods for leveraging LVLMs in multimodal recommendation contexts, providing valuable insights and guidance for future research and practical implementation. MSRBench sets the stage for further exploration and innovation in this field, aiming to advance the development of more accurate and personalized recommendation systems.

**Limitations and Future Work:** This work focuses on evaluating different strategies for applying LVLMs in multimodal sequential recommendation. However, due to resource limitations, we did not explore the potential impact of fine-tuning these models, which we plan to address in future research. Additionally, we recognize the importance of dataset diversity and aim to incorporate more varied datasets in subsequent studies.

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

# A Appendix

## A.1 Ethics

**General Ethical Conduct.** Our work follows standard ethical guidelines in AI and machine learning research. We clearly state our motivations, methodology, and results, ensuring transparency and reproducibility. The dataset used, Amazon Review Plus, is an extension of the publicly available Amazon Review dataset, and the data augmentation is conducted using LVLMs to generate additional item descriptions. The use of this data respects user privacy as the dataset does not include personally identifiable information. The primary aim of our proposed MSRBench and the Amazon Review Plus dataset is to explore improved strategies for integrating LVLMs into multimodal recommendation systems, potentially benefiting users by delivering more accurate and personalized recommendations.

**Potential Negative Societal Impacts.** While our work aims to enhance recommendation systems, it is crucial to consider the potential negative societal impacts. One significant concern is the reinforcement of existing biases present in the training data. LVLMs, trained on vast amounts of internet data, can inadvertently learn and propagate biases related to gender, race, and other sensitive attributes. When integrated into recommendation systems, these biases could lead to unfair or discriminatory recommendations, affecting user experience and perpetuating stereotypes. While our work does not directly cause negative societal impacts, it is crucial to take appropriate precautions to mitigate these potential effects.

## A.2 Prompts

The integration of LVLMs into multimodal recommendation systems is facilitated through a series of carefully designed prompts. Figure 6 shows examples of the prompts we use. Each color in the prompt structure highlights a specific component of the recommendation process, ensuring that the model can accurately interpret the information and perform the task.

**Blue** text represents the item titles that are passed to the LVLM. These are essential for identifying the products that the user has interacted with or is likely to be interested in.

**Yellow** text indicates the candidate item list, which is the pre-ranked list of items that the LVLM needs to process or rerank. This list is generated based on the user's previous interaction history and passed to the model for further refinement.

**Green** text refers to the generated image descriptions. These are automatically created by the LVLM based on the images of the items. These descriptions enrich the multimodal nature of the data by providing textual representations of visual elements.

**Red** text highlights the output format constraints. These instructions guide the LVLM on how to structure its output, ensuring that the recommendations are generated in a specific format that is easy to interpret and integrate into the system.

**Purple** text is used exclusively for the reranking process, where the LVLM is instructed to refine the pre-ranked list based on the likelihood of the user purchasing each item.

**Table 3: The hallucination rate (%) of three LVLMs across four datasets.**

| Model | Beauty | Clothing | Sports | Toys |
|---|---|---|---|---|
| Claude | 0.55 | 0.88 | **0.69** | 1.06 |
| GPT-4V | 1.33 | 1.22 | 1.15 | 1.52 |
| GPT-4o | **0.31** | **0.45** | 0.88 | **1.00** |

## A.3 Hallucination

Similar to the application of Large Language Models (LLMs) in recommendation systems, LVLMs for multimodal sequential recommendations also face the issue of hallucination, where the recommended item may not be in the valid candidate list. This section highlights the hallucination problem across various LVLMs. As shown in Table 3, the hallucination rates are below 2% across all datasets and models, indicating that the issue is generally not severe. Moreover, GPT-4o, which exhibits the strongest capabilities in recommendation tasks, also demonstrated the lowest hallucination rates.

## A.4 Analysis of Image Captions in the Amazon Review Plus Dataset

**Distribution of token length.** Figure 7 presents the distribution of token lengths for image captions generated by three different LVLMs (Claude-3-Opus, GPT-4V, GPT-4o) across four categories: beauty, clothing, sports, and toys. A consistent pattern is observed across all categories: Claude-3-Opus consistently generates longer and more detailed captions, indicated by higher mean token lengths and more concentrated distributions. In contrast, GPT-4V produces the shortest captions on average, with more spread-out distributions and significantly lower means. GPT-4o falls between the two, generating captions that are shorter than those of Claude-3-Opus but longer and less variable than those of GPT-4V. These findings suggest that Claude-3-Opus is more verbose and detailed in its captioning approach, GPT-4V is more concise, and GPT-4o strikes a balance between verbosity and conciseness. This highlights the varying strengths of each model in generating image captions, potentially influencing their suitability for different applications based on the required level of detail and verbosity.

**Word Cloud.** Furthermore, Figure 8 visualizes the word clouds for image captions generated by the three LVLMs across the four categories. Each row corresponds to a specific category, with columns

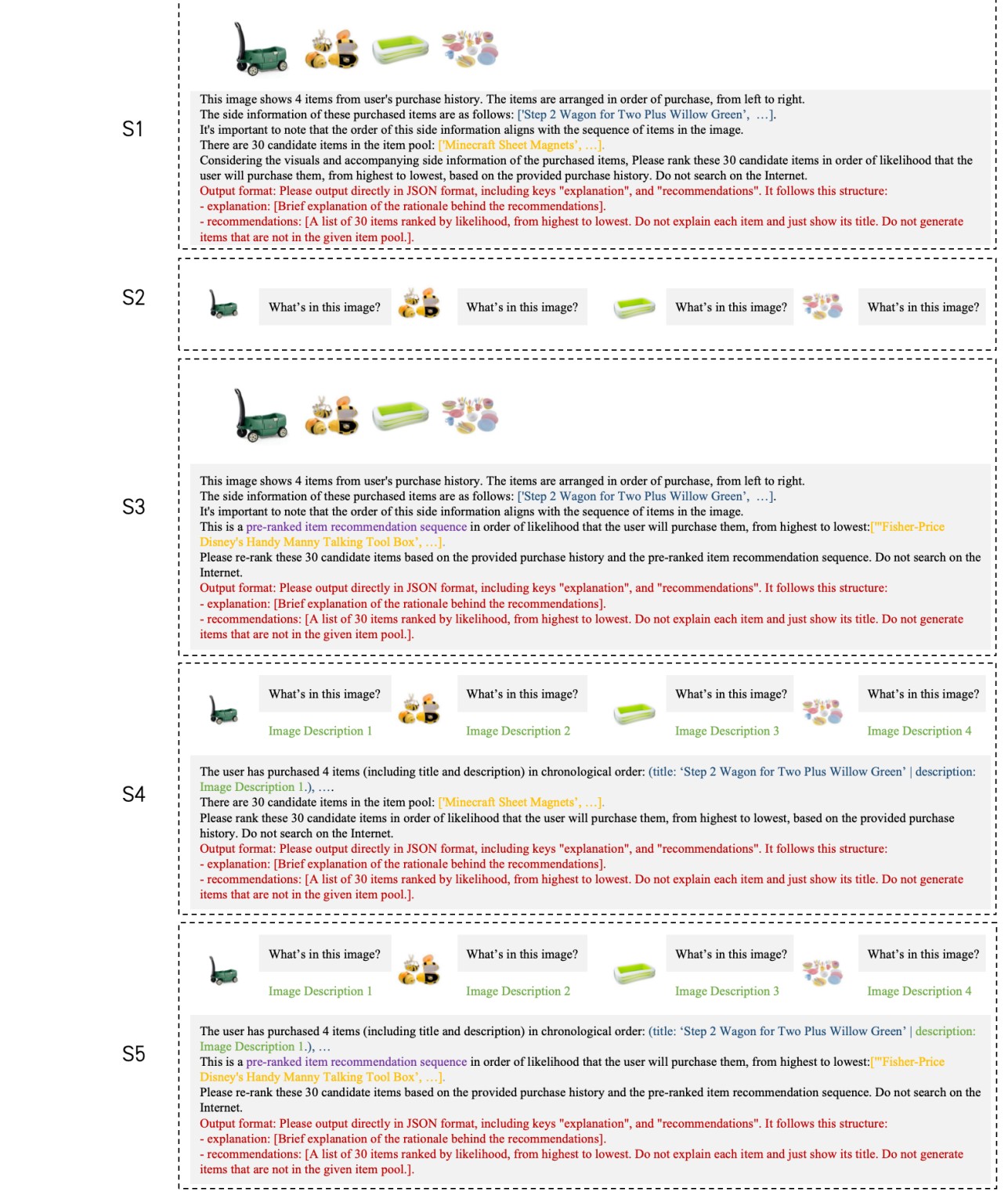

Figure 6: Examples of prompts used for different integration strategies. Blue, yellow, red and green texts represent item titles, candidate item lists, output format constraints, generated image descriptions, respectively. Purple text is only used for reranking.

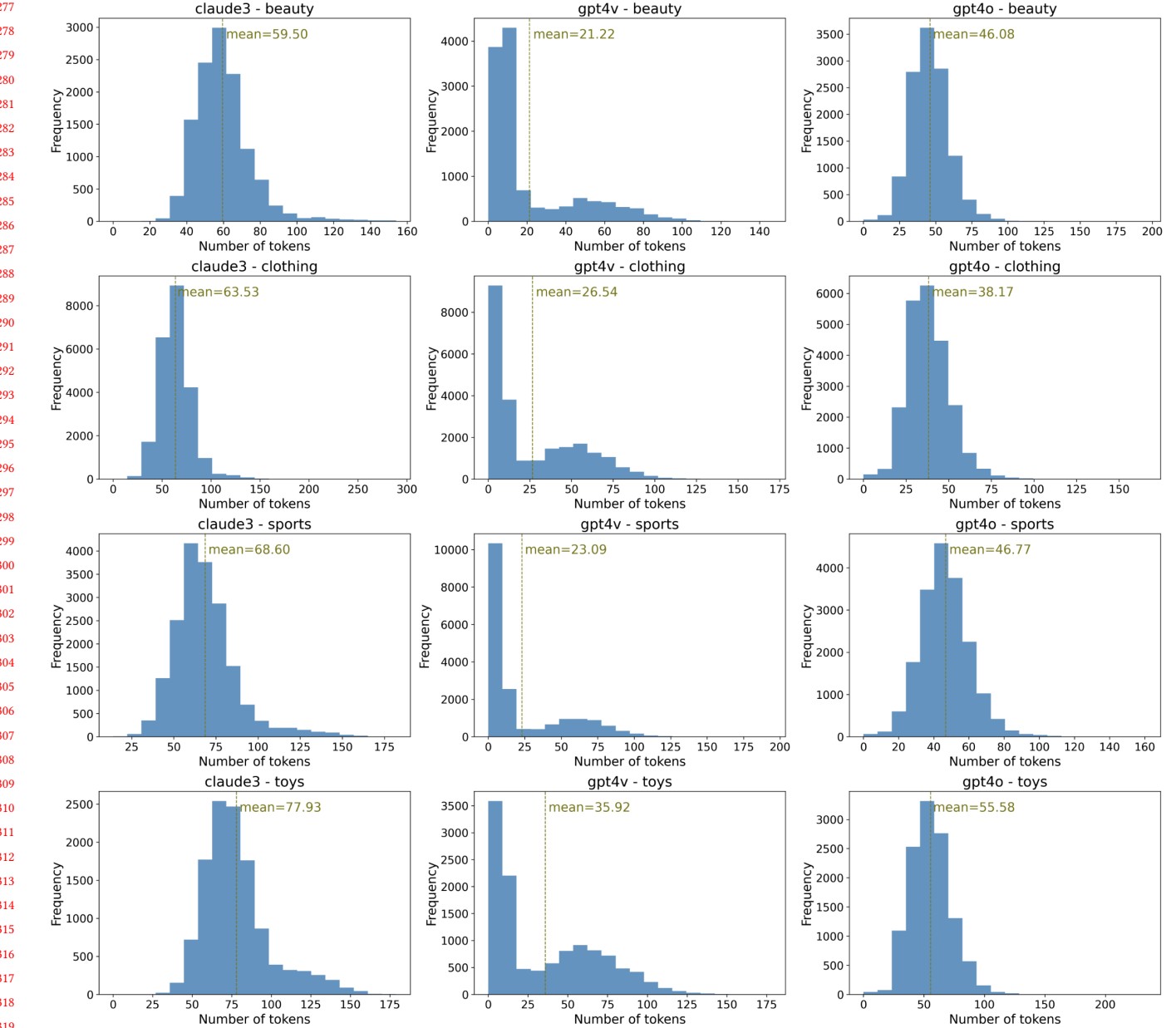

**Figure 7: Distribution of token lengths for image captions generated by three different LVLMs across four categories. The mean token length indicated by the dotted line.**

representing the different LVLMs. The size of each word indicates its frequency within the generated captions, highlighting the common terms and themes prevalent in the descriptions produced by each model. For instance, in the beauty category, frequent terms include "bottle," "hair," "color," and "nail." In the toys category, common words are "toy," "game," "set," and "children." This visualization effectively demonstrates the linguistic patterns and focal points of each model's captioning capabilities.

**Human Evaluation.** The human evaluation involved assessing the quality of image captions generated by three LVLMs across four distinct categories. Two evaluators (Eva_1 and Eva_2) participated in ranking the captions based on their quality. The evaluation metrics included the top1_ratio and the average position (avg_position), detail of which is showed in Table 5. The *top1_ratio* indicates the percentage of times a model's generated caption was ranked first by the evaluators. A higher top1_ratio signifies that the model

**Table 4: Detailed statistics of the four categories in the Amazon Review Plus dataset.**

| Dataset | Beauty | Sports | Clothing | Toys |
|---|---|---|---|---|
| #Users | 22,363 | 35,598 | 39,387 | 19,412 |
| #Items | 12,101 | 18,357 | 23,033 | 11,924 |
| #Photos | 12,023 | 17,943 | 22,299 | 11,895 |
| #Reviews | 198,502 | 296,337 | 278,677 | 167,597 |
| Sparsity (%) | 99.93 | 99.95 | 99.97 | 99.93 |
| Avg. Caption Len. (Claude-3-opus) | 59.49 | 68.59 | 63.52 | 77.93 |
| Avg. Caption Len. (GPT4-v) | 21.21 | 23.08 | 26.53 | 35.91 |
| Avg. Caption Len. (GPT4-o) | 46.08 | 46.76 | 38.16 | 55.57 |

frequently produced the best caption compared to others. The *average position (avg_position)* reflects the average rank of a model's captions, where a lower average position indicates better overall performance, as the model's captions were ranked higher (closer to first place) on average. The results indicate that Claude-3-Opus consistently outperformed the other models across all categories, achieving the highest top1_ratio and the best average position in every dataset. Specifically, Claude-3-Opus had an overall top1_ratio of 72.5% and an average position of 1.33, demonstrating its superior performance. In contrast, GPT-4o and GPT-4V lagged behind, with overall top1_ratios of 18.8% and 8.8%, respectively, and average positions of 2.04 and 2.63. These findings highlight the effectiveness of Claude-3-Opus in generating high-quality image captions, significantly surpassing the capabilities of GPT-4o and GPT-4V.

## A.5 Clarification on S2: LVLM as Item Enhancer

We do not adopt the strategy of converting textual modality into visual modality due to several significant challenges. One of the main concerns is the introduction of noise. Text-to-image generation, particularly at scale, often produces unreliable or low-quality visual representations. Moreover, the textual data we use, such as product titles, is often semantically abstract and includes specific brand names and concise descriptions that are difficult to accurately convert into meaningful images. As a result, generating images from these abstract texts likely results in visuals that lack the necessary detail and relevance to be useful in a recommendation context. Given these challenges, text-to-image conversion is not a practical solution for our study.

## A.6 More Implementation Details

For MoRec, we employ `bert-base-uncased` [9] as the text encoder and `clip-vit-base-patch32` [36] as the vision encoder. In MoRec(T) and MoRec(I), the output from the respective modality encoder is passed through a linear layer to generate the item representation. In the case of MoRec(T+I), the outputs from both the text encoder (BERT) and vision encoder (ViT) are concatenated and passed through a linear layer to form the final item vector. The dimensionality of the vectors from the text encoder is 768, while the vision encoder produces vectors of 512 dimensions. The final item vector, used by both SASRec and MoRec, is reduced to 512 dimensions. During training, the weights of BERT and ViT remain frozen to avoid overfitting.

For IISAN, we use a batch size of 64 and a maximum sequence length of 10. The learning rates are set as follows: 0.0005 for the

Beauty category, 0.0002 for Sports, 0.0003 for Toys, and 0.0005 for Clothing. All other training hyperparameters are consistent with those outlined in the main text.

For MMGCN, the regularization weight and learning rate are set to (0.1, 0.001) for Beauty, (0.0, 1e-4) for Clothing, (0.01, 5e-4) for Sports, and (1e-5, 0.001) for Toys. For FREEDOM, the learning rate is uniformly set to 0.001 across all datasets, with the regularization weight and dropout rate configured as follows: (0.9, 0.0001) for Beauty, (0.9, 0.0) for Clothing, (0.9, 0.0) for Sports, and (0.8, 0.0) for Toys. For BM3, we select a single GCN layer for all datasets, with the regularization weight and dropout rate set to (0.01, 0.3) for Beauty, (0.01, 0.5) for Clothing, and (0.1, 0.5) for both Sports and Toys.

The versions of the LVLMs we used are as follows: GPT-4o (`gpt-4o-2024-05-13`), GPT-4V (`gpt-4-vision-preview`), and Claude 3-Opus (`aws_claude3_sdk_opus`). Considering the substantial cost of API calls across the full test set, we follow the approach of prior works [8, 10, 13] by randomly sampling users from each category for evaluation. However, a critical question arises: *how many samples are sufficient to ensure representativeness?* To address this, we use **Cochran's Modified Formula for Finite Populations** [6, 18, 20], balancing representativeness and computational efficiency. Specifically, we select a 95% confidence level and a ±5% margin of error, which is a commonly accepted standard in similar studies, providing a good trade-off between statistical accuracy and the high costs associated with API calls. The formula is as follows:

$$n = \frac{N \cdot Z^2 \cdot p \cdot (1-p)}{e^2 \cdot (N-1) + Z^2 \cdot p \cdot (1-p)},$$

where $n$ is the sample size, $N$ the population size, $Z$ the Z-value for 95% confidence (1.96), $p$ the sample proportion (0.5), and $e$ the margin of error (0.05). For each category, the population sizes ($N$) are as follows: 22,363 for Beauty, 35,598 for Sports, 39,387 for Clothing, and 19,412 for Toys. Based on these values, we calculated the required sample sizes: approximately 378 for Beauty, 380 for Sports, 380 for Clothing, and 377 for Toys. To ensure consistency and representation, we sample 400 users per category. We validate this sampling approach on the Beauty dataset by comparing the recommendation performance trends between the 400 sampled users and the full dataset. Using metrics such as NDCG@5 and HR@5, we find that the difference between the full dataset and the sample results is less than 1%. This small difference is statistically insignificant and does not affect the overall trends or conclusions of our study, making the sample size of 400 users a reliable and computationally efficient choice.

## A.7 Dataset documentation and intended uses

The following questions are copied from "Datasheets for Datasets" [14].

### A.7.1 Motivation.

- **For what purpose was the dataset created?** Was there a specific task in mind? Was there a specific gap that needed to be filled? Please provide a description.
  The dataset was created to enhance the richness and comprehensiveness of the original Amazon Review Dataset by incorporating LVLM-generated image descriptions, thereby

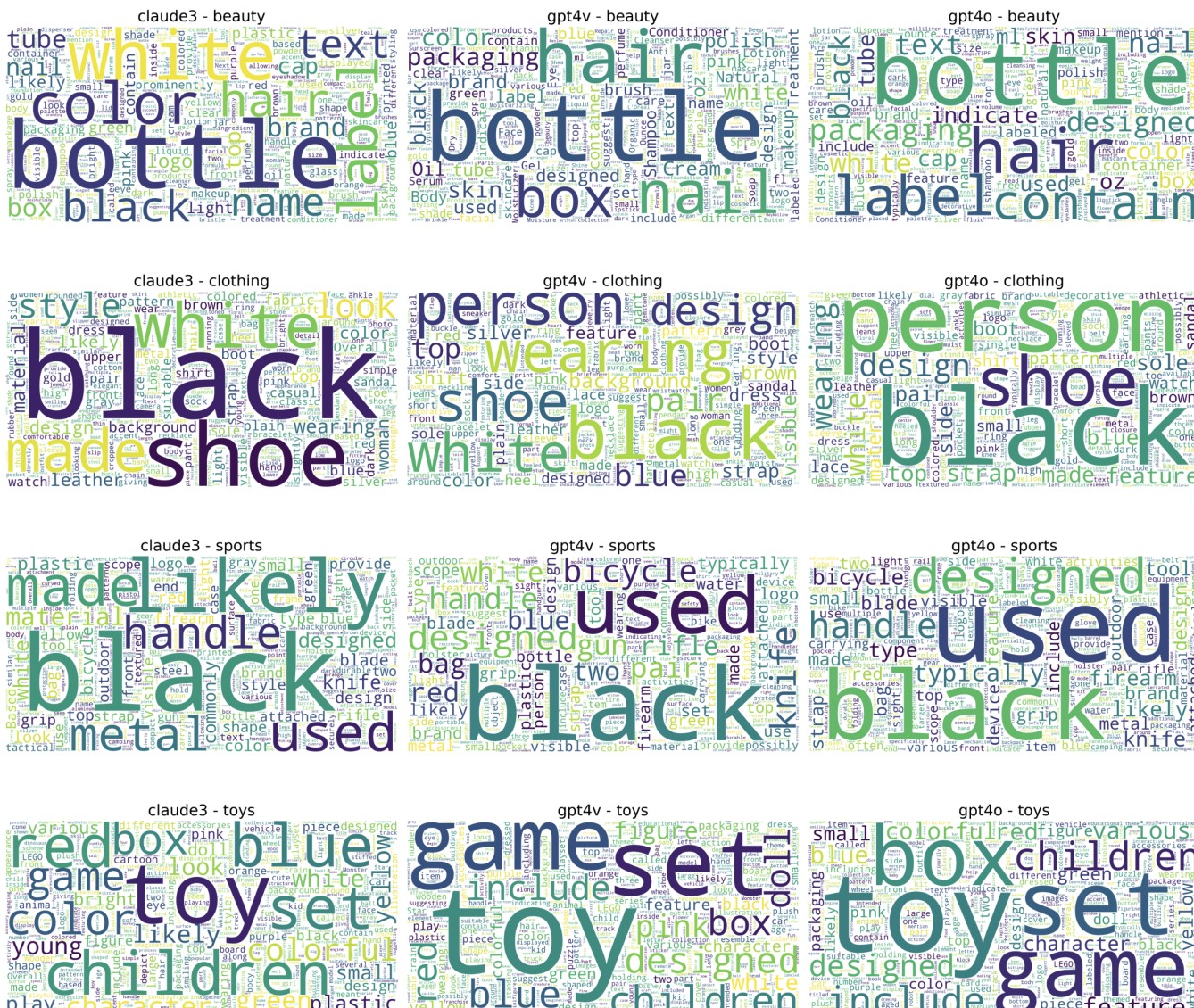

**Figure 8: Word cloud for image captions generated by three different LVLMs across four categories. The size of each word represents its frequency, highlighting common terms and themes generated by the models.**

addressing the gap of missing textual descriptions for products with only image data. The addition of image descriptions can help recommendation systems better understand product appearance features, potentially improving recommendation accuracy and user satisfaction, particularly for product categories where image descriptions are crucial (e.g., clothing and toys). Additionally, generating image description texts reduces computational complexity, as directly processing image data requires substantial computational resources and time. This transformation of image information into easily processed textual data improves both training and inference efficiency. Furthermore, in cross-modal retrieval tasks (e.g., finding corresponding images

from text descriptions or related text information from images), image description texts serve as a bridge, enhancing the accuracy and practicality of retrieval systems.

- **Who created the dataset (e.g., which team, research group) and on behalf of which entity (e.g., company, institution, organization)?**
  The Amazon Review Plus dataset was created by a collaborative research team consisting of members from multiple institutions and organizations.
- **Who funded the creation of the dataset?** If there is an associated grant, please provide the name of the grantor and the grant name and number.

**Table 5: Human evaluation of generated image captions from three LVLMs across four amazon review plus categories.**

| Dataset | Methods | Eva_1 | Eva_2 | avg_top1_ratio | avg_position |
|---|---|---|---|---|---|
| Beauty | Claude-3-Opus | 70.0% | 35.0% | **52.5%** | **1.52** |
| | GPT-4o | 30.0% | 60.0% | 45.0% | 1.68 |
| | GPT-4V | 0.0% | 5.0% | 2.5% | 2.80 |
| Toys | Claude-3-Opus | 80.0% | 60.0% | **70.0%** | **1.45** |
| | GPT-4o | 15.0% | 20.0% | 17.5% | 2.05 |
| | GPT-4V | 5.0% | 20.0% | 12.5% | 2.50 |
| Sports | Claude-3-Opus | 95.0% | 70.0% | **82.5%** | **1.20** |
| | GPT-4o | 5.0% | 10.0% | 7.5% | 2.17 |
| | GPT-4V | 0.0% | 20.0% | 10.0% | 2.62 |
| Clothing | Claude-3-Opus | 90.0% | 80.0% | **85.0%** | **1.15** |
| | GPT-4o | 0.0% | 10.0% | 5.0% | 2.25 |
| | GPT-4V | 10.0% | 10.0% | 10.0% | 2.60 |
| Overall | Claude-3-Opus | 83.8% | 61.3% | **72.5%** | **1.33** |
| | GPT-4o | 12.5% | 25.0% | 18.8% | 2.04 |
| | GPT-4V | 3.8% | 13.8% | 8.8% | 2.63 |

No funding was provided for this study, and it was conducted purely out of academic interest.

*A.7.2  Composition.*

- **What do the instances that comprise the dataset represent (e.g., documents, photos, people, countries)?** Are there multiple types of instances (e.g., movies, users, and ratings; people and interactions between them; nodes and edges)? Please provide a description.
  The Amazon Review Plus dataset builds upon the Amazon Review dataset by adding a new attribute: product image captions. These captions provide visual context and help in understanding the product's appearance and features as described by the powerful LVLMs.
  The original Amazon Review dataset includes user IDs, item IDs, item side information, and interactions between users and items such as ratings and reviews. Readers can check the details of this dataset in Amazon Review.
- **How many instances are there in total (of each type, if appropriate)?**
  The dataset comprises several types of instances across four categories: Beauty, Sports, Clothing, and Toys. Specifically, the dataset includes a total of 22,363 users for Beauty, 35,598 users for Sports, 39,387 users for Clothing, and 19,412 users for Toys. The number of items is 12,101 for Beauty, 18,357 for Sports, 23,033 for Clothing, and 11,924 for Toys.
  In terms of images, there are 12,023 photos for Beauty, 17,943 for Sports, 22,299 for Clothing, and 11,895 for Toys. The dataset also contains 198,502 reviews for Beauty, 296,337 for Sports, 278,677 for Clothing, and 167,597 for Toys.
  It is important to highlight that we generated captions for all items that include images, providing visual context and aiding in the understanding of the products' appearances and features.

- **Does the dataset contain all possible instances or is it a sample (not necessarily random) of instances from a larger set?** If the dataset is a sample, then what is the larger set? Is the sample representative of the larger set (e.g., geographic coverage)? If so, please describe how this representativeness was validated/verified. If it is not representative of the larger set, please describe why not (e.g., to cover a more diverse range of instances, because instances were withheld or unavailable).
  The Amazon Review Plus dataset contain four categories: Beauty, Sports, Clothing, and Toys. These four categories are widely adopted in existing recommendation research. Detailed statistics and descriptions of these datasets can be found in Section 4.2 of our submitted manuscript.
- **What data does each instance consist of?** "Raw" data (e.g., unprocessed text or images) or features? In either case, please provide a description.
  Each instance in the Amazon Review Plus dataset consists of several components. The reviews provide textual feedback from customers, while the image captions, generated for all items with images, offer visual context and help in understanding the product's appearance and features. Additionally, each instance includes structured features such as user ID, item ID, ratings, and review timestamps. These elements collectively enable comprehensive analysis of user interactions and product characteristics.
- **Is there a label or target associated with each instance?** If so, please provide a description.
  For each user interaction sequence, the last item in the sequence is used as the target.
- **Is any information missing from individual instances?** If so, please provide a description, explaining why this information is missing (e.g., because it was unavailable). This

does not include intentionally removed information, but might include, e.g., redacted text.

Some instances may have missing image captions. This is because certain products do not have associated images available at the time of data collection. As a result, these products lack the visual context provided by image captions

- **Are relationships between individual instances made explicit (e.g., users' movie ratings, social network links)?** If so, please describe how these relationships are made explicit.

  Yes, relationships between individual instances are made explicit through user IDs and item IDs. For example, a user's interaction with multiple items is linked by their user ID, and each item's reviews and ratings can be aggregated through their item ID. These relationships allow for analysis of user behavior and item popularity.

- **Are there recommended data splits (e.g., training, development/validation, testing)?** If so, please provide a description of these splits, explaining the rationale behind them.

  Yes, we recommend using leave-one-out strategy for data splitting. In this approach, for each user interaction sequence, the last item is used for testing, the second-to-last item is used for validation, and the remaining items are used for training.

- **Are there any errors, sources of noise, or redundancies in the dataset?** If so, please provide a description.

  Yes, the image captions generated by LVLMs might not always be accurate and could contain hallucinations or incorrect descriptions, leading to potential noise in the dataset.

- **Is the dataset self-contained, or does it link to or otherwise rely on external resources (e.g., websites, tweets, other datasets)?** If it links to or relies on external resources, a) are there guarantees that they will exist, and remain constant, over time; b) are there official archival versions of the complete dataset (i.e., including the external resources as they existed at the time the dataset was created); c) are there any restrictions (e.g., licenses, fees) associated with any of the external resources that might apply to a dataset consumer? Please provide descriptions of all external resources and any restrictions associated with them, as well as links or other access points, as appropriate.

  The dataset is primarily self-contained, but it may include links to product pages on Amazon for additional context. There are no guarantees that these external links will remain constant over time, and there are no official archival versions of the complete dataset including these external resources. Access to product pages is subject to Amazon's terms of service and availability.

- **Does the dataset contain data that might be considered confidential (e.g., data that is protected by legal privilege or by doctor–patient confidentiality, data that includes the content of individuals' non-public communications)?** If so, please provide a description.

No, the dataset does not contain data that might be considered confidential. All data included in the dataset is publicly available information from Amazon reviews.

- **Does the dataset contain data that, if viewed directly, might be offensive, insulting, threatening, or might otherwise cause anxiety?** If so, please describe why.

  The dataset may contain user-generated content that could be offensive, insulting, or otherwise cause anxiety. This includes reviews that may have negative or harsh language, as the content is not filtered or moderated for offensive language before being included in the dataset.

If the dataset does not relate to people, you may skip the remaining questions in this section.

- **Does the dataset identify any subpopulations (e.g., by age, gender)?** If so, please describe how these subpopulations are identified and provide a description of their respective distributions within the dataset.

  The dataset does not explicitly identify any subpopulations such as age or gender. All the data in the dataset is anonymized and does not include demographic information about the users. Therefore, subpopulations are not identified or described within the dataset.

- **Is it possible to identify individuals (i.e., one or more natural persons), either directly or indirectly (i.e., in combination with other data) from the dataset?** If so, please describe how.

  No, it is not possible to identify individuals directly or indirectly from the dataset. All personal identifiers are anonymized, and no additional data is provided that could be used in combination to identify users.

- **Does the dataset contain data that might be considered sensitive in any way (e.g., data that reveals race or ethnic origins, sexual orientations, religious beliefs, political opinions or union memberships, or locations; financial or health data; biometric or genetic data; forms of government identification, such as social security numbers; criminal history)?** If so, please provide a description.

  No, the dataset does not contain any data that might be considered sensitive. The data consists solely of user reviews, ratings, and related product information, without including any sensitive personal information such as race, ethnicity, sexual orientation, religious beliefs, political opinions, financial or health data, biometric or genetic data, government identification, or criminal history.

### A.7.3 Collection Process.

- **How was the data associated with each instance acquired?** Was the data directly observable (e.g., raw text, movie ratings), reported by subjects (e.g., survey responses), or indirectly inferred/derived from other data (e.g., part-of-speech tags, model-based guesses for age or language)? If the data was reported by subjects or indirectly inferred/derived from other data, was the data validated/verified? If so, please describe how.

The data associated with each instance was directly observable, consisting of raw text in the form of user reviews and ratings from the Amazon platform. Image captions were generated using LVLMs based on the available product images. The reviews and ratings are provided by users on Amazon.

- **What mechanisms or procedures were used to collect the data (e.g., hardware apparatuses or sensors, manual human curation, software programs, software APIs)?** How were these mechanisms or procedures validated?

  The image captions were generated using three SOTA LVLMs.

- **If the dataset is a sample from a larger set, what was the sampling strategy (e.g., deterministic, probabilistic with specific sampling probabilities)?**

  The dataset is a sample from a larger set of Amazon reviews. Specifically, we chose four widely adopted categories: Beauty, Sports, Clothing, and Toys.

- **Who was involved in the data collection process (e.g., students, crowdworkers, contractors) and how were they compensated (e.g., how much were crowdworkers paid)?**

  The Amazon Review Plus dataset was enhanced from the existing Amazon Review dataset. The enhancement process, which involved generating image captions using LVLMs, was carried out by engineers from Meituan. This work was done as a voluntary service, and therefore, no compensation was provided.

- **Over what timeframe was the data collected?** Does this timeframe match the creation timeframe of the data associated with the instances (e.g., recent crawl of old news articles)? If not, please describe the timeframe in which the data associated with the instances was created. The current dataset includes interactions spanning from May 1996 to October 2018.

- **Were any ethical review processes conducted (e.g., by an institutional review board)?** If so, please provide a description of these review processes, including the outcomes, as well as a link or other access point to any supporting documentation.

  All the data utilized in this research are publicly accessible and unrestricted. As the dataset does not contain any private or sensitive information, this research is exempt from ethical review.

If the dataset does not relate to people, you may skip the remaining questions in this section.

- **Did you collect the data from the individuals in question directly, or obtain it via third parties or other sources (e.g., websites)?**

  The data was obtained from third-party sources, specifically from the Amazon website, where users publicly post their reviews and ratings.

- **Were the individuals in question notified about the data collection?** If so, please describe (or show with screenshots or other information) how notice was provided, and provide a link or other access point to, or otherwise reproduce, the exact language of the notification itself.

  The individuals were not specifically notified about this particular data collection process. However, Amazon's terms of service inform users that their reviews and ratings may be publicly accessible and used for various purposes, including research.

- **Did the individuals in question consent to the collection and use of their data?** If so, please describe (or show with screenshots or other information) how consent was requested and provided, and provide a link or other access point to, or otherwise reproduce, the exact language to which the individuals consented.

  The individuals consented to the collection and use of their data through Amazon's terms of service, which users agree to when they post reviews and ratings on the platform. These terms specify that user-generated content is publicly accessible and can be used for research and other purposes.

- **If consent was obtained, were the consenting individuals provided with a mechanism to revoke their consent in the future or for certain uses?** If so, please provide a description, as well as a link or other access point to the mechanism (if appropriate).

  Users can revoke their consent by deleting their reviews or closing their accounts on Amazon, which removes their data from the platform. More details on how users can manage their data are provided in Amazon's privacy policy.

- **Has an analysis of the potential impact of the dataset and its use on data subjects (e.g., a data protection impact analysis) been conducted?** If so, please provide a description of this analysis, including the outcomes, as well as a link or other access point to any supporting documentation.

  An explicit data protection impact analysis was not conducted as part of this research. However, all data used are publicly available and do not contain private or sensitive information, minimizing potential negative impacts on data subjects. The research complies with standard ethical guidelines for using publicly accessible data.

### A.7.4 Preprocessing/cleaning/labeling.

- **Was any preprocessing/cleaning/labeling of the data done (e.g., discretization or bucketing, tokenization, part-of-speech tagging, SIFT feature extraction, removal of instances, processing of missing values)?** If so, please provide a description. If not, you may skip the remaining questions in this section.

  Yes, we handled missing values by either imputing or discarding incomplete instances. Additionally, image captions were generated using LVLMs to provide visual context for items with images.

- **Was the "raw" data saved in addition to the preprocessed/cleaned/labeled data (e.g., to support unanticipated future uses)?** If so, please provide a link or other access point to the "raw" data.

  No.

- **Is the software that was used to preprocess/clean/label the data** available? If so, please provide a link or other access point.
  No specific software was used; instead, custom scripts written in Python were utilized to interact with APIs from OpenAI and Anthropic.

### A.7.5   Uses.

- **Has the dataset been used for any tasks already?** If so, please provide a description.
  Yes, the original Amazon Review dataset has been widely used for various recommendation tasks, such as sequential recommendation, rating prediction, explainable recommendation, and more. Our enhanced version of the dataset is also suitable for these tasks, providing additional visual context through image captions.
- **What (other) tasks could the dataset be used for?**
  In addition to the multimodal recommendation tasks discussed in this paper, the dataset can be used for rating prediction, explainable recommendation, review summarization, cross-modal retrieval, and other related tasks.
- **Is there anything about the composition of the dataset or the way it was collected and preprocessed/cleaned/labeled that might impact future uses?** For example, is there anything that a dataset consumer might need to know to avoid uses that could result in unfair treatment of individuals or groups (e.g., stereotyping, quality of service issues) or other risks or harms (e.g., legal risks, financial harms)? If so, please provide a description. Is there anything a dataset consumer could do to mitigate these risks or harms?
  No.
- **Are there tasks for which the dataset should not be used?** If so, please provide a description.
  There are no specific tasks for which the dataset should not be used. However, users should ensure that the application of the dataset aligns with ethical guidelines and data usage policies.

### A.7.6   Distribution.

- **Will the dataset be distributed to third parties outside of the entity (e.g., company, institution, organization) on behalf of which the dataset was created?** If so, please provide a description.
  Yes, the dataset will be distributed to third parties outside of the entity.
- **How will the dataset will be distributed (e.g., tarball on website, API, GitHub)?** Does the dataset have a digital object identifier (DOI)?
  The dataset will be distributed via GitHub upon acceptance. It does not have a digital object identifier (DOI) at this time.
- **When will the dataset be distributed?**
  The dataset will be distributed before the conference.
- **Will the dataset be distributed under a copyright or other intellectual property (IP) license, and/or under applicable terms of use (ToU)?** If so, please describe this license and/or ToU, and provide a link or other access point

to, or otherwise reproduce, any relevant licensing terms or ToU, as well as any fees associated with these restrictions. The dataset will be distributed under the Apache License 2.0.
- **Have any third parties imposed IP-based or other restrictions on the data associated with the instances?** If so, please describe these restrictions, and provide a link or other access point to, or otherwise reproduce, any relevant licensing terms, as well as any fees associated with these restrictions.
  No.
- **Do any export controls or other regulatory restrictions apply to the dataset or to individual instances?** If so, please describe these restrictions, and provide a link or other access point to, or otherwise reproduce, any supporting documentation.
  No.

### A.7.7   Maintenance.

- **Who will be supporting/hosting/maintaining the dataset?** The first author of this paper, will be supporting, hosting, and maintaining the dataset.
- **How can the owner/curator/manager of the dataset be contacted (e.g., email address)?**
  The first author can be contacted via email.
- **Is there an erratum?** If so, please provide a link or other access point.
  No.
- **Will the dataset be updated (e.g., to correct labeling errors, add new instances, delete instances)?** If so, please describe how often, by whom, and how updates will be communicated to dataset consumers (e.g., mailing list, GitHub)?
  No, the dataset will not be updated regularly. If updates are planned in the future, the reasons will be elaborated on our GitHub repository.
- **If the dataset relates to people, are there applicable limits on the retention of the data associated with the instances (e.g., were the individuals in question told that their data would be retained for a fixed period of time and then deleted)?** If so, please describe these limits and explain how they will be enforced.
  Not applicable, as the dataset does not contain data that directly relates to identifiable individuals.
- **Will older versions of the dataset continue to be supported/hosted/maintained?** If so, please describe how. If not, please describe how its obsolescence will be communicated to dataset consumers.
  Yes, older versions of the dataset will continue to be supported, hosted, and maintained. If updates are made, the old version will be maintained, and the new version will be released with an updated version number, such as Amazon Review Plus 2.0.
- **If others want to extend/augment/build on/contribute to the dataset, is there a mechanism for them to do so?** If so, please provide a description. Will these contributions be validated/verified? If so, please describe how. If not, why not? Is there a process for communicating/distributing

these contributions to dataset consumers? If so, please provide a description.

Yes, if others want to contribute to the dataset, they can submit a pull request or contact us via email. Contributions will be validated and verified by the maintainers before being merged into the main dataset. This process ensures the quality and integrity of the dataset, and updates will be communicated through the GitHub repository.

## A.8   Accessibility

1. Links to access the dataset and its metadata will be made available upon acceptance.
2. The data is saved in a JSON format, with an example provided in the README file.
3. The dataset will be maintained on an official GitHub account by the authors.
4. The dataset will be released under the Apache License 2.0.

## A.9   Data Usage

The authors bear all responsibility in case of violation of rights.

