# OpenReview forum: "When Large Vision Language Models Meet Multimodal Sequential Recommendation: An Empirical Study"
_ACM.org/TheWebConf/2025/Conference — WWW 2025 Oral_

### Official Review · Reviewer_bhPt · 2024-11-29

**Novelty:** 4
**Technical Quality:** 5

**Review:**

Summary
The paper proposes an evaluation of five different strategies for integrating LVLMs in the multimodal sequential recommendation task. Moreover, the authors promise to release an augmented version of the Amazon dataset, called the Amazon Review Plus dataset, that they realized and utilized in the benchmark. The dataset contains image descriptions of products in the Amazon Review dataset.

Enumeration of weaknesses
•	The chosen LVLMs are all proprietary, it will no longer be possible to reproduce the results of the study once the models used are no longer available.
•	The discussion of the results for RQ1 could be more thorough.
•	The experiments on image input modes were performed on a single dataset (beauty) and a single model (GPT-4V), impairing the generalizability of the results shown for RQ2.

Enumeration of strengths
•	The work investigates the performance of different strategies to leverage proprietary LVLMs in multimodal sequential recommendation, this topic has never been thoroughly investigated before.
•	The experimental setting is detailed, and all choices are motivated.
•	The work provides extensive details about the baseline models.

Relevance
The paper is relevant to the RecSys track of WebConf

Novelty
The paper addresses the relevance of LVLMs in the multimodal sequential recommendation task. The proposed integration strategies are not novel, but the comprehensive comparison is interesting nonetheless. The dataset created by the authors could be useful for the Recommender Systems community.

Methodology
The authors propose to leverage proprietary LVLMs in the task of multimodal sequential recommendation. They design, implement and evaluate five different integration strategies to understand if and how the use of LVLM can improve recommendation performance.

Reproducibility
The authors promise to release the source code for the study and the dataset they collected and used. If they follow up on this promise, the work will be reproducible as long as the APIs for the LVLMs used are available.

Experimental Design + Evaluation
The experimental design follows the standard common practices related to the Amazon Reviews dataset in the sequential recommendation scenario, this includes 5-core filtering and the leave-one-out splitting strategy. The only difference with the common experimental setting is the use of 29 negative samples, but this choice is motivated by practical considerations.

**Questions:**

- Do you have any insights to give as to why the S3 strategy performs so much better than S1? Intuitively, one would think that it is easier to rank higher the ground truth item in a list containing 29 random negative samples than in a list containing 29 other items already chosen by a recommender system.
-Why did you choose not to also use the candidate items images in S1, S3, S4, S5?

**Reviewer Confidence:**

3: The reviewer is confident but not certain that the evaluation is correct

**Scope:**

2: The connection to the Web is incidental, e.g., use of Web data or API

---

### Official Review · Reviewer_eJYb · 2024-11-30

**Novelty:** 5
**Technical Quality:** 4

**Review:**

This paper introduces MSRBench, a comprehensive benchmark, and the Amazon Review Plus dataset, addressing the significant challenge of effectively integrating Large Vision Language Models (LVLMs) into multimodal sequential recommendation systems. The problem it tackles—how to leverage LVLMs for better alignment of visual and textual data in personalized recommendations—is both interesting and meaningful, given the increasing reliance on multimodal content in recommendation scenarios. By evaluating state-of-the-art LVLMs such as GPT-4o across five integration strategies, the paper provides a clear and valuable framework for analyzing their roles, supported by extensive experimental results. While the proposed framework is novel, it needs to be further generalized and be able to cover the relevant LVLM methods. More importantly, since the study’s scope is overly broad, its experimental setting/results need refinement to substantiate its claims fully.

Minor things
- In Abstract, mmendation -> recommendation
- Section 2 title, Mutimodal should be Multimodal
- In Section 4, investing? or investigating?
- Reference formatting could be improved.

**Questions:**

- Why are item descriptions used only for S4 and S5? At the very least, item titles should be included under the images so that they align correctly with the prompt.
- Is "What's in this image?" the only prompt used in S2?
- In my humble opinion, different LVLM backbone models specialize in different roles. Are there any ablation studies comparing their effects on various components?
- It cannot be guaranteed that the same prompt works equally effectively across different LVLM backbones. Have you observed any differences in this regard?
- Among the many SOTA approaches leveraging LVLM introduced in this paper, why is only [44] compared?
- The framework does not seem applicable for a strict sequential recommendation setting, as a long prompt cannot be used consistently throughout an entire conversation. How can this limitation be addressed?
- Figure 2 appears redundant with Table 1.
- It would be beneficial to include results for k=10 for H@k and N@k, as this is a conventional setting.
- While extracting item descriptions, using a more precise prompt might help achieve higher-quality descriptions. Do you have any relevant observations or results regarding this?
- Does each item have only one image? Even items within the same category may have different camera angles, which could influence performance. Could you provide any insights on this?
- Why does GPT4o perform comparably to the item-only case? How do you treat this in the main result table?

**Reviewer Confidence:**

3: The reviewer is confident but not certain that the evaluation is correct

**Scope:**

3: The work is somewhat relevant to the Web and to the track, and is of narrow interest to a sub-community

---

### Official Review · Reviewer_hDat · 2024-12-02

**Novelty:** 6
**Technical Quality:** 5

**Review:**

This paper provides an empirical study on integrating the Large vision language models (LVLMs) into sequence recommendation. The author considers five different variants for using the LVLM, e.g., as Recommender, Item Enhancer, and Reranker. The author augments the existing dataset, i.e., Amazon Review, with image captioning and benchmarks different LVLM and state-of-the-art multi-modality sequence recommendation models. The empirical findings show that leveraging LVLM as Reranker brings large improvement to existing models.

**Strength":

1. To the best of my knowledge, this should be the first work on benchmarking different usage of LVLM for sequence recommendation. The empirical study is comprehensive, relevant, and timely. The reviewer believes this work will largely promote the study of LVLM for the community's sequence recommendation.

2. The design of the MSRBench is reasonable and sound. The five strategies considered in the MSRBench are relevant. The chosen methods for comparison are comprehensive and cover different aspects, making the benchmark meaningful. The final takeaway message is clear.

**Weakness**:

1. Although the empirical study of MSRBench is comprehensive, the insights and analysis behind the empirical observation are lacking. For example, the author shows that leveraging LVLM as a reranker brings the largest improvement in Section 3.4. However, there is no interpretation of the underlying reasons for these performance changes. The author needs to consider the following questions to help them reason about this improvement:

(a) compared to other strategies, what is the advantage of using LVLM as a Reranker?

(b) Why reranker using LVLM can improve the result? What about using other non-LVLM methods to do the reranker? What is the advantage of LVLM compared to other non-LVLM methods for re-ranking?

2. The author describes the prompt design for MSRBench but does not provide the ablation study on the prompt design. The author may want to study how the prompt change may influence the benchmark results.

3. The author stated in lines 316-319 that open-source models like Qwen-VL and GLM-4v exhibited poor instruction-following capabilities, and thus, they are excluded from the comparison. The reviewer believes that even the failure of these models is worth reporting since it can inspire future work to improve the open-source LVLM. Moreover, the author should also consider the Chameleon [a] and CM3leon [b] from Meta for comparison.

[a] Chameleon: Mixed-modal early-fusion foundation models. ArXiv 2024

[b] Scaling autoregressive multi-modal models: Pretraining and instruction tuning. ArXiv 2023

**Questions:**

Please refer to the Review section for details.

**Reviewer Confidence:**

4: The reviewer is certain that the evaluation is correct and very familiar with the relevant literature

**Scope:**

4: The work is relevant to the Web and to the track, and is of broad interest to the community

---

### Official Review · Reviewer_qY6e · 2024-12-03

**Novelty:** 4
**Technical Quality:** 2

**Review:**

### Summary
This paper explores the integration of Large Vision Language Models (LVLMs) in multimodal sequential recommendation systems. Specifically, the authors introduce MSRBench, a comprehensive benchmark designed to evaluate different LVLM integration strategies in sequential recommendation scenarios, which includes three state-of-the-art LVLMs (namely, GPT-4Vision, GPT-4o, and Claude-3-Opus). Then, the authors find that using LVLMs as rerankers is the most effective approach and GPT-4o consistently achieves the best performance across most scenarios.

---

### Strengths
* The authors perform extensive experiments with diverse model designs for LVLM-powered recommendation tasks.
* This paper is well-written and easy to follow.

---

### Weaknesses
* It is questionable whether the proposed setup of considering the multimodality (text + image) is truly needed for sequential recommendation tasks, since considering multimodality seems not helpful over the text-only modality models (as shown in Figure 3), i.e., it may not be necessary to consider the vision-modality when we tackle sequential recommendation tasks.
* To handle multiple (previously interacted) images, the authors simply concatenate them into one image; however, it is questionable whether the LVLMs can capture meaningful information from this concatenated image. For instance, if the number of images is large (e.g., 100), each image within the concatenated image may not be processed well as it occupies only a very small portion of the entire image (e.g., 1%).
* The experiments are performed only with proprietary models, which limits the broad extensibility of this work (as researchers who are not able to run experiments with those models may not benefit from this work). In addition to this, while the authors mention that they tried some open-source LVLMs, I am wondering whether the poor capabilities of those models are due to their sizes and whether very recent LVLMs (such as Llama) are still not able to perform the target tasks.

**Questions:**

For the conventional method called MoRec, it is beneficial to consider image modality alongside the text modality; however, this benefit of using image modality is not so clear when using recent (and probably larger) LVLMs. Can you explain the (potential) reason for this?

**Reviewer Confidence:**

3: The reviewer is confident but not certain that the evaluation is correct

**Scope:**

3: The work is somewhat relevant to the Web and to the track, and is of narrow interest to a sub-community